# A Next-Generation Adenoviral Vaccine Elicits Mucosal and Systemic Immunogenicity and Reduces Viral Shedding after SARS-CoV-2 Challenge in Nonhuman Primates

**DOI:** 10.3390/vaccines12020132

**Published:** 2024-01-27

**Authors:** Sarah N. Tedjakusuma, Colin A. Lester, Elena D. Neuhaus, Emery G. Dora, Samanta Gutierrez, Molly R. Braun, Sean N. Tucker, Becca A. Flitter

**Affiliations:** Vaxart, Inc., South San Francisco, CA 94080, USA; sarahtedja@gmail.com (S.N.T.); clester@vaxart.com (C.A.L.); eneuhaus@vaxart.com (E.D.N.); edora@vaxart.com (E.G.D.); sgutierrez@vaxart.com (S.G.); mbraun@vaxart.com (M.R.B.); stucker@vaxart.com (S.N.T.)

**Keywords:** adenovirus vaccines, immunity, mucosal, SARS-CoV-2, immunoglobulin A

## Abstract

As new SARS-CoV-2 variants continue to emerge and impact communities worldwide, next-generation vaccines that enhance protective mucosal immunity may have a significant impact on productive infection and transmission. We have developed recombinant non-replicating adenovirus serotype 5 (rAd5) vaccines delivered by mucosal administration that express both target antigen and a novel molecular adjuvant within the same cell. Here, we describe the immunogenicity of three unique SARS-CoV-2 rAd5 vaccine candidates and their efficacy following viral challenge in non-human primates (NHPs). Intranasal immunization with rAd5 vaccines expressing Wuhan, or Beta variant spike alone, or Wuhan spike and nucleocapsid elicited strong antigen-specific serum IgG and IgA with neutralizing activity against multiple variants of concern (VOC). Robust cross-reactive mucosal IgA was detected after a single administration of rAd5, which showed strong neutralizing activity against multiple VOC. Additionally, mucosal rAd5 vaccination increased spike-specific IFN-γ producing circulating T-cells. Upon Beta variant SARS-CoV-2 challenge, all the vaccinated NHPs exhibited significant reductions in viral load and infectious particle shedding in both the nasal passages and lower airways. These findings demonstrate that mucosal rAd5 immunization is highly immunogenic, confers protective cross-reactive antibody responses in the circulation and mucosa, and reduces viral load and shedding after SARS-CoV-2 challenge.

## 1. Introduction

Although currently approved parenteral vaccines against severe acute respiratory syndrome coronavirus 2 (SARS-CoV-2) reduce severe disease and hospitalization, these injectable formulations induce minimal protective mucosal immunity where pathogen entry occurs [1,2,3]. As SARS-CoV-2 variants of concern (VOC) with increasing fitness emerge, vaccine-induced serum antibody responses are becoming less effective and increasingly short-lived [4,5,6]. Ideally, next-generation SARS-CoV-2 vaccines will induce both systemic and mucosal immunity, elicit cross-reactive responses against emerging variants, and have the capacity to reduce community transmission.

A primary function of mucosal immunoglobin A (IgA) is to prevent microorganisms from entering host tissues at mucosal sites. IgA is secreted as a multimeric protein, which allows one antibody molecule to bind multiple particles [7]. The increased IgA valency confers superior neutralizing activity against invading pathogens compared to immunoglobin G (IgG) [8,9], and prevents respiratory and enteric pathogens from infecting mucosal sites by immune exclusion, steric hindrance, and the inhibition of intracellular viral production [10,11,12]. Dimeric IgA has been shown to be 15 times more neutralizing than IgG against SARS-CoV-2 [13,14]. Therefore, generating potent IgA responses to SARS-CoV-2 antigens through mucosal vaccination may be key for blocking pathogen entry, and reducing community transmission.

Non-replicating human adenovirus serotype 5 (rAd5) oral vaccines delivers target antigens alongside a dsRNA adjuvant directly to the mucosa. This rAd5 vaccine platform has been previously successfully used in human clinical trials for norovirus and influenza virus indications [15]. Oral rAd5 immunization delivers target antigens expressed intracellularly or on the cellular surface of host cells. A molecular dsRNA adjuvant in the oral rAd5 vaccine platform is used to overcome oral tolerance in the small intestine, provoking innate signaling cascades that subsequently stimulate strong adaptive responses. In humans, oral rAd5 vaccination generates target-antigen-specific mucosal IgA and has demonstrated efficacy against influenza virus challenge [16]. Oral rAd5 vaccinations have been administered to over 700 subjects, have an excellent safety profile, and generate robust humoral and cellular immune responses against targeted antigens [17,18]. Local and distal mucosal IgA responses are induced in the gut, the oral cavity, and the nose [17]. Orally administered rAd5 is not hindered by pre-existing adenoviral antibodies in humans [18], whereas intramuscular delivery can be significantly hindered by pre-existing immunity. Furthermore, the potency of this mucosal immunization strategy and its ability to impact transmission has been demonstrated via aerosol delivery in a pre-clinical hamster SARS-CoV-2 study [19].

SARS-CoV-2 studies have utilized nonhuman primates (NHPs), including African green monkeys (AGMs), to investigate the immunogenicity of vaccine candidates and their ability to reduce virus replication/shedding in the respiratory tract [20]. We previously tested immunogenicity of ED88, ED90, and ED94 in Cynomolgus macaques [21]. At the time of this study, there was a shortage of macaques, a favored NHP model organisms used for testing SARS-CoV-2 vaccine efficacy. AGMs have been previously shown to be a suitable model for both SARS-CoV-1 and SARS-CoV-2 infection, eliciting both antibody and T-cell responses after infection [22,23,24,25]. 

The objective of this study was to select a rAd5 SARS-CoV-2 clinical candidate for advancement by assessing immunogenicity and efficacy in NHPs, in addition to identifying a vaccine candidate that generated robust mucosal and systemic immunity to multiple VOC while conferring protection against infectious viral challenge. We also examined if a variant-specific immunization approach provided greater protective efficacy, compared to a heterologous vaccine antigen approach. We evaluated a two-dose mucosal immunization regimen of three rAd5 candidates, ED88 (Wuhan spike + nucleocapsid), ED90 (Wuhan spike), and ED94 (Beta spike), which encode the spike proteins alone or in tandem with nucleocapsid. Intranasal delivery was used as a proxy for oral delivery due to complexities involved in NHP oral tablet delivery. We demonstrate the induction of protective systemic and mucosal immunity following rAd5 mucosal immunization, resulting in a significant reduction in infectious viral loads in both the upper and lower airways of AGMs upon SARS-CoV-2 Beta variant challenge, which was the prominent circulating variant at the time of this study.

## 2. Materials and Methods

### 2.1. Adenoviral Vaccine Constructs

Adenovirus rAd5 vaccine plasmid constructs were created based on available DNA sequences of SARS-CoV-2 in Genbank Accession No. MN90847.3 for the parental Wuhan strain and Global Initiative on Sharing All Influenza Data Accession Number EPI_ISL_678597 for the Beta strain. The published amino acid sequences of the SARS-CoV-2 spike or nucleocapsid proteins were used to generate human codon-optimized nucleotide sequences. The nucleotide sequences were cloned into E1/E3 deficient rAd5 and purified as previously described [16,17].

### 2.2. Ethics Statement

This study was conducted in compliance with the Public Health Service Policy on Human Care and Use of Laboratory Animals using a protocol (protocol number 21-097 approved July 2021) approved by the Institutional Animal Care and Use Committee (IACUC). The facility where this research was conducted is accredited by the Association for Assessment and Accreditation of Laboratory Animal Care (AAALAC International) and conducted in accordance with the 8th edition of the Guide for the Care and Use of Laboratory animals. The total number of animals used, group size, and the number of groups in this study were considered a minimum required to characterize the immunogenicity and efficacy of the vaccine candidates. NHPs were observed twice daily and physically examined for nasal discharges, respiratory rate, respiratory character, body condition score, hydration status, pulse oximetry levels, and food consumption (biscuits eaten). No animals were excluded for poor health during the course of the study.

### 2.3. Study Design, Immunization, and Challenge

Twenty-seven adult NHPs of the AGM species *Chlorocebus sabaeus* (52% Male/48% Female) originating in St. Kitts were received from PrimGen (Hines, IL, USA). AGMs were randomly assigned to five groups (Appendix A) by the contract research organization BioQual (Rockville, MD, USA), while controlling for sex and weight. All animals were provided free access to water by a water bottle and fed ad libitum. The diet consisted of certified Purina Monkey Diet (cat. no. 5038, Purina, St. Louis, MO, USA). Environmental controls for the animal room were set to maintain 64 °F to 84 °F, a relative humidity of 30–70%, and a 12 h light/12 h dark cycle. Visual, auditory, and olfactory contact with conspecifics, food variety, and treat items were offered at multiple feeding times each day. Climbing devices, manipulanda, and foraging devices were all utilized to provide an enhanced environment and to promote psychological well-being.

Animals were intranasally immunized with rAd5 5 × 10^10^ infectious units on day 0 and day 28 using a mucosal atomization device, with 0.1 mL administered per nostril. Three groups were administered either ED88, ED90, or ED94 for both prime and boost, and a fourth group was given an intramuscular injection of purified spike protein, NR-52308n, from BEI Resources (Manassas, VA, USA) followed by an intranasal boost of ED94.

Prior to immunization, serum was collected from each animal on days −1, 28, 42, and 54. Bronchoalveolar lavage fluid (BALF) was collected prior to challenge on day 54 and on days 2, 5, and 8 post challenge. Nasal secretions were collected from both nostrils using 3 mm synthetic absorptive matrix (SAM) Nasosorption FX-i devices (cat# NSFL-FXI-15 swabs, Mucosal Diagnostics), on days 0, 28, and 54 and were immediately frozen and stored at −80 °C.

On day 63, all animals were challenged with SARS-CoV-2 Beta variant (isolate h-CoV-19/South Africa/KRISP-K005325/2020) by administering a 5 × 10^4^/mL tissue culture infection dose 50 (TCID_50_) by intranasal and intratracheal routes for a total of 1 × 10^5^ TCID_50_ administered to each animal. Post challenge, animals were monitored daily for any abnormal clinical observations. To quantify viral loads and shedding, FLOQSwab nasal swabs (Copan Diagnostics, Murrieta, CA, USA) were collected on days 1, 2, 5, and 8 post challenge. BALF was collected on day 54 post vaccination and days 2, 5, and 8 post challenge.

### 2.4. Serum and Mucosal Antibody Responses to SARS-CoV-2 Variants

Serum, nasal, and BALF antibody responses to trimerized spike protein and receptor binding domain (RBD) protein were quantified against Wuhan and VOC using V-PLEX SARS-CoV-2 panel 7 (Meso Scale Diagnostics [MSD], Rockville, MD, USA) and U-PLEX plates (MSD, Rockville, MD, USA). Serum was diluted 1:1000 in 1% Enhanced Chemiluminescence (ECL) Blocking Agent (Cytiva, Marlborough, MA, USA) in 1X phosphate-buffered saline (PBS) with 0.05% Tween 20 and added to MSD plates. To measure IgA in the nasal secretions, IgA was eluted from SAM devices and diluted 1:2 and 1:4 in 1% ECL Blocking Agent Sample. Sample incubation times and IgG or IgA antibody detection were conducted per manufacturer instructions. Data were acquired using the MSD Sector Imager 120 instrument. Antigen-specific nasal IgA was normalized to the total amount of IgA in the corresponding sample. Fold change was calculated by dividing the normalized antigen-specific nasal IgA values at each timepoint by the day −1 values.

### 2.5. Nasal and BALF Angiotensin-Converting Enzyme-2 (ACE-2) Functional Antibody Response by Surrogate Virus Neutralization Test (sVNT)

Functional antibody responses in the nasal cavity and lower airways were measured by surrogate virus neutralization tests (sVNTs) using the MSD platform. Mucosal samples were diluted 1:2 and 1:4 in diluent (diluent 100, MSD, MD, USA) prior to addition to Coronavirus Plate 2 (MSD, MD, USA) or U-Plex plate containing Delta, Beta, and Omicron RBD proteins. Following a one-hour incubation, angiotensin-converting enzyme-2 (ACE-2) Sulfo-Tag protein was added to the wells and incubated for an additional hour before developing and reading on an MSD Sector Imager 120 instrument. Percent inhibition of ACE-2 via the SARS-CoV-2 antigen was calculated according to manufacturer instructions by subtracting the ratio of sample relative light units (RLUs) to the lowest calibration standard RLU from one and multiplying by 100.

### 2.6. Serum Neutralizing Antibodies by Plaque Reduction Neutralization Test (PRNT)

Plaque reduction neutralization test (PRNT) assays were conducted at BioQual as previously described [26]. Unknown heat-inactivated serum samples were serially diluted 3-fold and incubated with 30 pfu/well of virus (hCoV-19/South Africa/KRISP-K005325/2020) at 37 °C, 5.0% CO_2,_ for 1 h. The serum and virus mixture was added in duplicate at 175,000 cells/well to Vero E6 cells (ATCC, cat# CRL-1586) in 24-well plates, and incubated at 37 °C, 5.0% CO_2,_ for 1 h. A total of 1 mL of warm 0.5% methylcellulose media was added, and the plates were incubated at 37 °C, 5% CO_2,_ for 3 days. After 0.2% crystal violet staining (20% MeOH, 80% dH_2_O) and washing with dH_2_O, the plaques in each well were recorded and the IC_50_ and IC_90_ titers were calculated based on the average number of plaques detected in the virus control wells.

### 2.7. Spike-Specific T-Cell Secretion of IFN-γ by ELISpot

IFN-γ-secreting cells were measured by enzyme-linked immunosorbent spot (ELISpot) assay according to manufacturer instructions (cat# CT121-PR5, U-CyTech, Utrecht, The Netherlands) after being stimulated for 24 hours with overlapping SARS-CoV-2 spike 15-mer peptide pools (cat# 130-127-951, Miltenyi Biotec, Bergisch Gladbach, Germany). Plates were developed and spot forming units (SFUs) were counted and normalized to spots/million cells (ZellNet Consulting, Fort Lee, NJ, USA).

### 2.8. Viral Load and Shedding

Viral load was quantified by measuring the amount of RNA copies per mL using a qRT-PCR assay, as previously described [26]. Viral RNA was isolated from swabs or BALF using the Qiagen MinElute virus spin kit (cat. no. 57704). Subgenomic RNA (sgRNA) was isolated using primers and a probe specifically designed to amplify and bind to a region of the nucleocapsid gene messenger RNA from SARS-CoV-2. The signal was compared to a known standard curve of plasmid containing the sequence of part of the messenger RNA, including part that was not in the virus, and calculated to give copies per gram of tissue. Genomic RNA (gRNA) was isolated using primers and a probe specifically designed to amplify and bind to a conserved region of nucleocapsid gene. The signal was compared to a known standard curve and calculated to give copies per mL. Infectious viral particles were measured by TCID_50_ assay using Vero TMPRSS2 cells, as previously described [26]. For samples with fewer than three positive wells with cytopathic effect (CPE), the TCID_50_ value could not be calculated using the Reed–Muench formula.

### 2.9. Statistical Analysis

All statistical analyses were performed using GraphPad Prism (GraphPad Software, v10, San Diego, CA, USA). The methods used for determining statistical significance were two-way analysis of variance (ANOVA) tests with Tukey–Kramer’s post hoc test, and cross-correlation Spearman’s r matrix tests using two-tailed analysis.

## 3. Results

### 3.1. Mucosal Vaccination Elicits Strong Cross-Reactive Systemic Immunity

We evaluated three rAd5 candidate vaccines that express SARS-CoV-2 antigens: ED88 (Wuhan spike + nucleocapsid protein), ED90 (Wuhan spike), and ED94 (Beta spike) (Figure 1A,B). The two transgenes of ED88 are under the control of cytomegalovirus (CMV) promoters and b-actin promoters, respectively, whereas the single spike transgene of ED90 and ED94 are under the control of a CMV promoter. The beta actin promoter was chosen to drive the expression of the SARS-CoV-2 nucleocapsid transgene in ED88 to minimize the inclusion of repetitive sequences in this construct.

The animals were immunized with a prime–boost regimen on day 0 and day 28, while nasal secretions and serum were collected on days −1, 28, 42, and 54 (Figure 1C). Three groups received two intranasal vaccine doses of either ED88, ED90, or ED94. Additionally, one group received an unadjuvanted intramuscular injection prime with recombinant Wuhan spike protein followed by an intranasal boost immunization with ED94 (Figure 1D). The purpose of this last group was to compare the immunological response of rAd5 to a different vaccination approach. At the time of this study, parenterally administered mRNA and adenovirus vaccines were under emergency use authorization and not available for research studies. A control group of five animals was intranasally administered PBS without vaccine candidates. On day 63, the animals were challenged with 1 × 10^5^ TCID_50_ SARS-CoV-2 Beta variant, which was circulating at the time of the challenge. Serial nasal and BALF samples were collected to determine the viral load and shedding.

The vaccine serum IgG responses elicited by the vaccination groups were quantified using MSD technology to trimerized spike proteins from Wuhan, Beta, Delta, and Omicron (BA.1) variants (Figure 1E). Following a single vaccine dose, we observed ED90 promoted the strongest serum IgG responses, with a 100-fold increase to the Wuhan and Delta and 10-fold against the Beta and Omicron variant antigens. The boost administration of ED90 did not result in further enhanced serum IgG; however, elevated antibody levels were maintained through day 54. Higher cross-reactive serum IgG responses against VOC spike protein and RBD were observed following two doses of ED88 compared to ED90. ED94 had strong serum IgG responses to the vaccine-matched Beta spike and RBD proteins after a single dose, which increased following the boost vaccination. The NHPs that were initially vaccinated intramuscularly with recombinant spike protein exhibited low serum IgG responses which increased after intranasal boost immunization with ED94, suggesting that rAd5 vaccination was responsible for promoting humoral responses against spike and RBD.

Animals immunized with matched vaccine antigens generated slightly higher IgG responses against the corresponding full-length spike and RBD proteins. However, ED94-vaccinated animals elicited lower cross-reactive serum IgG responses to other variants compared to animals immunized with Wuhan-based ED90 and ED88. Of note, all the vaccinated groups elicited similar serum IgA responses against the Wuhan, Beta, Delta, and Omicron spike and RBD proteins (Appendix A). Although the ED88 vaccine was designed to express the nucleocapsid protein antigen intracellularly, we did not detect N-specific serum antibody (data not shown). These results suggest that rAd5 intranasal administration generated a high magnitude of IgG and IgA serum responses to matched antigens, while the Wuhan-based ED90 and ED88 promoted the strongest circulating antibody breadth to multiple VOC.

### 3.2. Robust Cross-Reactive Nasal IgA Is Elicited Following Mucosal Vaccination

We next measured antigen-specific mucosal IgA against Wuhan, Beta, Delta, and Omicron spike or RBD in nasal secretions (Figure 2). Antigen-specific IgA was normalized to total IgA in the sample and shown as fold change compared to day −1. Prime immunization with ED90 elicited IgA increases of 600-fold to Wuhan and over 200-fold to Beta, Delta, and Omicron (BA.1). After ED90 boost, the quantity of spike-binding nasal IgA further increased to 3000-fold above baseline. A single administration of ED88 induced a slightly lower antigen-specific nasal IgA response to the Wuhan, Beta, Delta, and Omicron spike and RBD proteins compared to ED90. Upon boost administration, ED88 elicited similar levels of cross-reactive IgA in nasal secretions compared to ED90 immunized AGMs. The ED94-vaccinated animals generated slightly higher nasal IgA responses to the matching Beta spike and RBD proteins after a single dose. Interestingly, upon the boost immunization of ED94, the cross-reactive nasal IgA response to other variants was similar in magnitude to those observed in the ED88- and ED90-primed and -boosted animals. The animals vaccinated with spike protein by intramuscular injection did not produce nasal IgA on day 28; however, following the boost with ED94, spike-specific IgA mucosal responses were observed in the nasal secretions at day 54. These results show mucosal immunization with ED88, ED90, and ED94 rAd5 induced robust increases in cross-reactive IgA to multiple VOC in the nasal mucosa.

### 3.3. Boost Immunization Enhances Neutralizing Antibody Activity in Both the Peripheral and Mucosal Compartments

Next, we investigated if intranasal rAd5 immunization induced functional serum antibody responses using an sVNT assay. The ED90 vaccination induced strong levels of neutralizing serum antibodies against the Wuhan and Delta RBD proteins by day 28 (Figure 3A). The ED88 vaccination induced increased levels of neutralizing serum antibodies against the Wuhan, Beta, and Delta RBD proteins after boost vaccination. Following a single dose of ED94, low levels of cross-neutralizing serum antibodies against the Wuhan and Delta RBDs were generated compared to ED88 and ED90. Boosting with ED94 elicited stronger serum neutralizing responses against the homologous Beta RBD compared to ED88 and ED90. The animals vaccinated with purified spike protein by intramuscular injection did not produce serum neutralizing antibodies by day 28; however, a subsequent single boost of intranasal ED94 did increase serum antibody functional activity to Beta by day 54. While ED90 and ED88 elicited cross-reactive antibodies against Omicron BA.1 in the serum, these responses did not prevent RBD:ACE-2 interactions found in the sVNT.

In parallel, we conducted PRNT assays using day 54 serum against the SARS-CoV-2 Beta variant (RSA) in a Biosafety Level-3 facility. The serum PRNT verified our internal sVNT results (Figure 3A) as demonstrated by other laboratories [27,28], and further confirmed that the animals administered two intranasal doses of ED94 had the strongest serum neutralizing activity to the homologous Beta variant (Figure 3B). All the animals in the ED94 group had IC_50_ titers above the proposed protective levels in NHP models [26]. Notably, 83% of the animals primed and boosted with ED88 and 67% of the ED90-immunized animals also had protective serum IC_50_ titers, suggesting that mucosal immunization with ancestral Wuhan spike induces cross-reactive serum antibodies (Figure 3B). This result corroborated our earlier findings in a Cynomolgus macaque model, that the mucosal administration of ED90 generates cross-reactive functional antibodies [21].

We further evaluated the functional neutralizing activity in mucosal secretions using an sVNT assay. We found that two intranasal doses of ED88, ED90, or ED94 were required to induce potent neutralizing activity in the nasal mucosa (Figure 3C). The prime and boost of ED88 and ED90 elicited strong cross-reactive nasal neutralizing antibodies to multiple VOC. Two doses of ED94 also generated nasal blocking antibodies to Wuhan and Beta, but less to Delta. The intramuscular prime administration of recombinant spike protein followed by an intranasal ED94 boost weakly generated mucosal antibody neutralizing activity. This observation further indicated two doses of rAd5 were needed to enhance the neutralizing antibodies. While we observed increases in antibody binding to the Omicron BA.1 spike in the nasal samples in the ED88, ED90, and ED94 vaccinated animals, these responses were lower against RBD (Figure 2) and conferred weak or no neutralizing activity (Figure 3C). The antibody functional activity was also quantified by sVNT in BALF on day 54 (Appendix A); however, compared to the nasal cavity, less neutralization was observed against all the VOC in the BALF. Overall, these results show that the prime and boost mucosal administration of the ED88, ED90, and ED94 rAd5 vaccine candidates elicited the most cross-reactive functional neutralizing antibodies in both the serum and mucosal compartments to multiple VOC.

### 3.4. Mucosal Administration of rAd5 Vaccines Elicits Antigen-Specific T-cells

To evaluate the vaccine-induced spike-specific IFN-γ producing T cells was measured on days 0, 7, 28, and 42 by ELISpot and cells secreting IFN-γ were enumerated as SFUs and expressed as SFU/million peripheral blood mononuclear cells (PBMCs) (Figure 4A) or the fold change over day 0 (Figure 4B). All the groups vaccinated with ED88, ED90, and ED94 elicited spike-specific T-cell responses by day 28. The IFN-γ-secreting T-cell responses continued to rise two weeks after the boost vaccination (day 42) in the ED88- and ED94-vaccinated groups. The intramuscular vaccination with spike protein failed to elicit T-cell responses; however, following the administration of ED94, the T-cell responses were similar to the groups vaccinated with ED88 or ED94. Lastly, there were no significant differences in the IFN-γ-secreting T-cells with the inclusion of nucleocapsid protein or between the four vaccinated groups at day 42.

### 3.5. Viral Replication and Shedding Is Significantly Reduced in Immunized Animals after Challenge

To determine whether mucosal immunization reduces the viral load following challenge, the viral RNA was measured against the conserved regions of the nucleocapsid gene in the nasal samples and BALF by RT-qPCR. Twenty-four hours post challenge, all the vaccinated groups had considerably lower total viral transcripts measured by gRNA, and reduced viral replication measured by sgRNA compared to the unvaccinated control group (Figure 5A). Of note, the viral RNA continued to decrease over time in the nasal passages at two, five, and eight days post challenge in all the animals vaccinated with rAd5. Five days post challenge, the sgRNA levels fell below the limit of detection in the ED94-immunized animals, indicating the homologous vaccine antigen improved the viral clearance. In the lower airways, all the groups had high viral loads at two days post challenge in the BALF, which rapidly decreased by four logs at eight days post challenge (Figure 5B). Interestingly, the animals immunized with purified spike followed by ED94 intranasal vaccination had the lowest levels of viral RNA in the nasal passages, but the highest level of transcripts in the lower airways. This suggests that two doses of intranasal ED94 are required to substantially reduce viral replication in the lower airways. These data show that mucosal immunization with either the matched or mismatched rAd5 vaccines effectively reduced the viral load in the nasal passages and lower airways.

To determine if vaccination with ED88, ED90, and ED94 could reduce infectious viral shedding, the mucosal samples were quantified by TCID_50_ from the nasal swabs and BALF. The high levels of shedding observed in the nasal mucosa in the unvaccinated control animals peaked at two days post challenge and was detectable at eight days post challenge (Figure 6A). In contrast, the infectious virus significantly decreased in all the vaccinated animals 24 h after the challenge compared to the control group. Notably, all the immunization regimens reduced the amount of infectious viral shedding below the lower limits of quantification (LLOQ) in most of the samples. While the four animals administered spike protein intramuscularly followed by a single dose of ED94 had reduced mucosal and systemic humoral immunogenicity compared to the other immunized groups, these animals were still protected against the matched challenge Beta variant and had low levels of shedding in the nose, suggesting that a single administration of ED94 was effective in reducing shedding in the upper airways. In contrast to the nasal mucosa, the infectious virus particles detected in the lower airways were somewhat reduced in the immunized animals, with only the AGMs vaccinated with two doses of ED94 having significant decreases in shedding in the lower airways (Figure 6B). While the immunizations with the mismatched vaccine antigens were cross-protective in the nasal mucosa, vaccination with the matched antigen variant, ED94, was more effective at reducing shedding in the lower airways upon challenge.

Lastly, we used a Spearman cross-correlation matrix to explore the relationship between the mucosal IgA responses detected in the upper and lower airway with the viral loads in individual animals. We compared the nasal IgA concentrations two weeks post boost administration on day 54, to the infectious viral load and gRNA two days post challenge (Figure 7A). Increases in the nasal IgA quantity against Wuhan or Beta negatively correlated with increased Beta variant viral gRNA and shedding, demonstrating the importance of mucosal IgA in protection. We also evaluated the BALF IgA concentrations and found similar trends (Figure 7B). This analysis further suggests that antigen-specific IgA in the mucosa promotes a reduction in the infectious viral load and shedding in the upper and lower airway.

## 4. Discussion

Our results show that the mucosal rAd5 SARS-CoV-2 vaccines were highly immunogenic and elicited functional antibody responses in both the circulation and mucosal surfaces, corroborating earlier research [21]. We observed that all the rAd5 SARS-CoV-2 candidates evaluated generated robust mucosal and systemic cross-reactive antibody responses to mismatched antigens. While all the SARS-CoV-2 rAd5 vaccine candidates induced mucosal IgA in the respiratory tract, we found that two intranasal vaccine doses were most effective for inducing potent cross-reactive neutralizing IgA. This was also observed in the serum neutralization responses, aligning with other NHP SARS-CoV-2 studies, where the serum PRNT responses peaked after two vaccine doses [21,29]. Although the binding antibodies trended slightly higher to full length spike compared to the RBD proteins in the animals immunized with the matching antigen, these cross-reactive antibodies were functionally protective in vivo against a mismatched viral challenge. Further, ED90 generated the highest magnitude and breadth of antibodies against all the VOC tested overall and was effective following challenge against a mismatched SARS-CoV-2 variant, similar to our previous findings [21]. We also observed a strong increase in T-cell spike-specific IFN-γ secretion following vaccination with all the rAd5 constructs. These immunogenic responses from all the vaccine candidates conferred protection in a SARS-CoV-2 Beta variant challenge and greatly reduced the viral load and shedding.

Developing vaccines that induce protective mucosal immunity may be superior at preventing infection and reducing the transmission of respiratory pathogens. In a guinea pig influenza infection study, injected dimeric IgA antibodies were detected in the nasal mucosa and prevented transmission from infected animals [30]. A recent study investigating intranasal or intratracheal boost vaccination with rAd26 found antibody and T-cell responses generated in the lung following immunization were the strongest immunologic correlate for protection against SARS-CoV-2 challenge in rhesus macaques [31]. In a hamster transmission study, oral and intranasal vaccination with ED90 followed by a high-dose infection of SARS-CoV-2 significantly reduced subsequent aerosol transmission to vaccine-naive hamsters. Further, elevated mucosal antibody levels of vaccinated hamsters was correlated with the reduction in transmission [19]. This study and the previously cited study demonstrate that mucosal rAd5 vaccine administration has the potential to reduce upper respiratory shedding and limit the transmission of infectious aerosols.

Although intranasal delivery has shown promising immunogenic responses and protection in preclinical studies against SARS-CoV-2, intranasal vaccine administration has been less impressive in human adults [32,33]. This has been shown with Flumist^®^, where the intranasal vaccine was more effective in young children than in older adults [34]. Intranasal delivery of rAd vaccines against SARS-CoV-2 in humans was unsuccessful in two different studies [35,36], and while an intranasal vaccine was approved for use in India, little information about its induced mucosal immunity is publicly available [37]. One possible explanation for the limited success of intranasal vaccination is a substantial amount of pre-existing cross-reactive IgA in the upper respiratory tract from prior encounters with endemic coronaviruses. These antibodies may hinder the recognition of recombinant spike protein by naive or memory B cells. In contrast, more consistent vaccine-induced mucosal immunogenicity has been observed in humans when rAd is delivered by aerosol, bypassing the upper respiratory tract [38], or by oral delivery to the intestine [16]. In this preclinical study, intranasal delivery was used as a proxy for oral delivery due to the added complexities involved in NHP tablet delivery. However, once the clinical candidate is selected, the oral tablet administration of the vaccine to the intestine will be used in humans. This rAd5 tablet-based immunization method previously showed efficacy in a human influenza challenge study [16], demonstrating the potential of this approach.

Injected SARS-CoV-2 vaccines have demonstrated protection from severe disease, but do not induce significant mucosal responses that block infection or inhibit transmission. A study in Singapore reported that vaccinated subjects were infected by the Delta variant at similar rates to unvaccinated subjects, though vaccinated subjects cleared the virus slightly faster [18]. A Danish study with a highly vaccinated population reported that household transmission still occurred 25–32% of the time even with a full vaccination regimen, including a booster [39]. These studies highlight that current vaccine regimens insufficiently prevent infection and viral transmission, having great social, medical, and economic impacts. The mucosal immunization of even a small portion of the population may reduce viral transmission dynamics. In an adolescent study, schools were randomly chosen to offer intranasal Flumist^®^ vaccine and compared to those where Flumist^®^ was not offered. Children and adults from intervention-school households had fewer medical visits for influenza-like illness, despite only 47% of the students being vaccinated in these schools [40]. Given the increase in nasal IgA associated with Flumist^®^ vaccination in children [41], these results highlight the potency of mucosal immunity in reducing transmission.

One limitation of this study was our use of spike protein as a positive control rather than a commercially available parentally administered SARS-CoV-2 vaccine. We were unable to obtain commercially produced vaccines for comparison to the rAd5 mucosal vaccine at the time this experiment was conducted. As mRNA and protein vaccines are becoming more available for use in basic research, we are currently further investigating differential responses to a variety of injected vaccines compared to oral or intranasally administered rAd5.

It has been three years since the Beta variant of SARS-CoV-2 was the dominant circulating variant. While multiple variants have arisen since the Beta variant was first detected, the results from this challenge study suggest that mucosal rAd5 SARS-CoV2 vaccines are effective and induce cross-reactive antibodies to heterologous variants. Other studies have investigated if a variant-matched vaccine is critical for SARS-CoV-2 protective immunity. Boost intramuscular immunization with rAd26 containing either Wuhan spike or Omicron BA.1 spike were compared in NHPs previously immunized with rAd26 Wuhan. While all the regimens induced strong antibody responses, the Omicron BA.1 booster offered only a moderate advantage in Omicron-specific immunity. Furthermore, while the neutralizing antibodies were shown to be crucial for protection, the T-cell responses also played a significant role in protection against the Omicron challenge. This study suggests that while matched vaccines offer slightly better protection against homologous SARS-CoV-2 variants, Wuhan-based vaccines continue to provide effective protection against variant infections [42]. In future studies, we will need to examine whether efficacy is maintained against new variants following vaccination with ED90, or if a variant-specific rAd5 vaccine will be a more protective approach.

## 5. Conclusions

Protection against continual waves of emerging SARS-CoV-2 variants with the current injectable vaccines may be difficult, due to waning immunity and the infrastructure needed to administer needle-based immunizations. The data presented here suggest that delivering rAd5 vaccines to a mucosal surface is an alternative immunization approach that can generate both serum and mucosal responses, while protecting against infection and reducing shedding. Ideally, next-generation SARS-CoV-2 vaccines will enhance mucosal responses and reduce community transmission, in addition to preventing severe disease. Advancing next-generation mucosal vaccines that are easy to administer, store, and distribute is paramount to vaccine equity and the effectiveness of global public health responses to future waves of SARS-CoV-2 infections.

## Figures and Tables

**Figure 1 vaccines-12-00132-f001:**
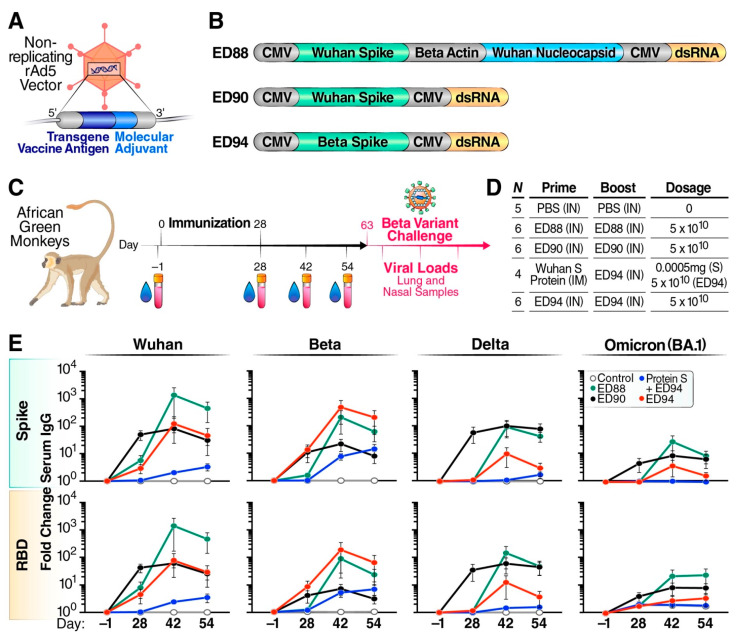
Mucosal immunization with ED88, ED90, and ED94 vaccine candidates generates cross-reactive serum IgG. (**A**) Illustration of non-replicating adenovirus serotype 5 (rAd5) vaccine engineered to deliver both transgene antigen and dsRNA molecular adjuvant to the same cell. (**B**) Schematic of transgene antigens in rAd5 SARS-CoV-2 vaccines ED88, ED90, and ED94. (**C**) Diagram of study design indicating vaccine administration schedule, serum and nasal swab sample collection, and infectious challenge with SARS-CoV-2 Beta variant. (**D**) Vaccines administered during prime and boost immunization, number of animals (*N*) assigned per group, route of administration, intranasal (IN) or intramuscular (IM), and dosage. (**E**) Serum spike-specific IgG was quantified by Meso Scale Discovery against Wuhan, Beta, Delta, and Omicron (BA.1) variants on days −1, 28, 42, and 54. Experimental groups included vehicle control (open white circles, *n* = 5), ED88 (green circles, *n* = 6), ED90 (black circles, *n* = 6), intramuscular delivery of spike protein followed with ED94 boost (blue circles, *n* = 4), ED94 (red circles, *n* = 6). Data normalized to total IgG in each sample timepoint and expressed as fold change from baseline at day −1, ±SEM; top row, full-length trimerized spike; bottom row, RBD. Abbreviations: CMV, cytomegalovirus promoter; dsRNA, double-stranded RNA; IN, intranasal administration; PBS, phosphate-buffered saline; RBD, receptor binding domain; rAd5, non-replicating adenovirus serotype 5; SEM, standard error of the mean.

**Figure 2 vaccines-12-00132-f002:**
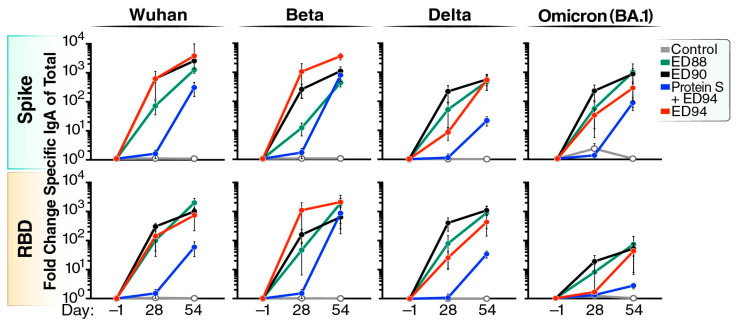
Mucosal immunization stimulates robust cross-reactive nasal IgA to multiple spike protein VOC. Spike-specific IgA against Wuhan, Beta, Delta, and Omicron (BA.1) and total IgA were quantified from nasal secretions collected at days −1, 28, and 54. Vehicle control (open white circles, *n* = 5), ED88 (green circles, *n* = 6), ED90 (black circles, *n* = 6), intramuscular delivery of spike protein followed with ED94 boost (blue circles, *n* = 4), ED94 (red circles, *n* = 6). The top row shows full-length trimerized spike results while the bottom row shows RBD protein results. Data are normalized to total IgA in each sample timepoint and expressed as fold change from baseline at day −1, ±SEM. Abbreviations: IgA, immunoglobulin A; RBD, receptor binding domain; SEM, standard error of the mean; VOC, variants of concern.

**Figure 3 vaccines-12-00132-f003:**
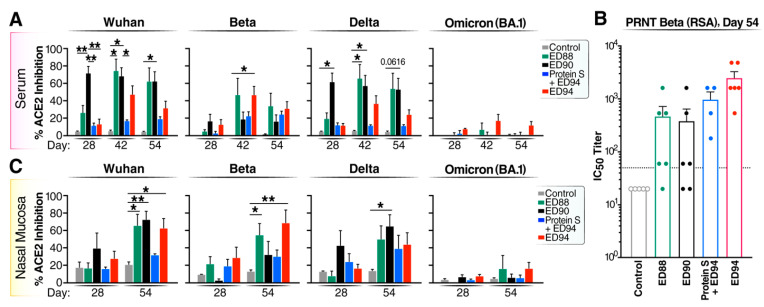
Mucosal vaccination induces neutralizing antibodies in serum and nasal secretions. (**A**) Serum neutralizing antibody activity against the RBD portions of Wuhan, Beta, Delta, and Omicron variants by sVNT. Percent ACE-2 inhibition shown on days 28, 42, and 54 for serum, vehicle control (grey bars, *n* = 5) ED88 (green bars, *n* = 6), ED90 (black bars, *n* = 6), intramuscular delivery of spike protein followed with ED94 boost (blue bars, *n* = 4), and ED94 (red bars, *n* = 6). Data expressed as mean ± SEM; two-way ANOVA with Tukey–Kramer post hoc test. (**B**) Serum neutralizing IC50 antibody titers per animal on day 54 by PRNT assay against SARS-CoV-2 Beta variant (first documented in RSA). The dotted line denotes proposed NHP IC50 protective serum titer (IC50 = 50). (**C**) sVNT measuring nasal mucosal antibodies percent ACE-2 inhibition at day 28 and day 54. Abbreviations: ACE-2, angiotensin-converting enzyme-2; ANOVA, analysis of variance; IC50, half maximal inhibitory concentration; NHP, nonhuman primate; PRNT, plaque reduction neutralization test; RSA, Republic of South Africa; SEM, standard error of the mean; sVNT, surrogate virus neutralization test.

**Figure 4 vaccines-12-00132-f004:**
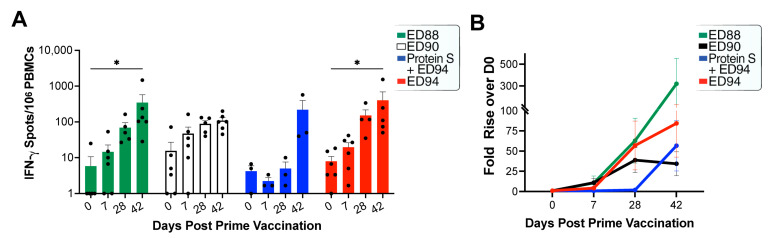
Immunization with rAd5 vaccine elicits spike specific T-cell responses. PBMCs were stimulated for 24 hours with spike protein 15-mer peptide pool. (**A**) IFN-γ secretion in animals vaccinated with vehicle control (grey, *n* = 5), ED88 (green, *n* = 6), ED90 (black, *n* = 6), primed on day 0 with intramuscular delivery of spike protein followed with ED94 boost on day 28 (blue, *n* = 4), and ED94 (red, *n* = 6). Bars represent mean IFN-γ secretion per million PBMCs ± SEM. (**B**) IFN-γ secretion per million PBMCs expressed as fold change over day 7. Abbreviations: IFN-γ, interferon-γ; PBMCs, peripheral blood mononuclear cells; SEM, standard error of the mean.

**Figure 5 vaccines-12-00132-f005:**
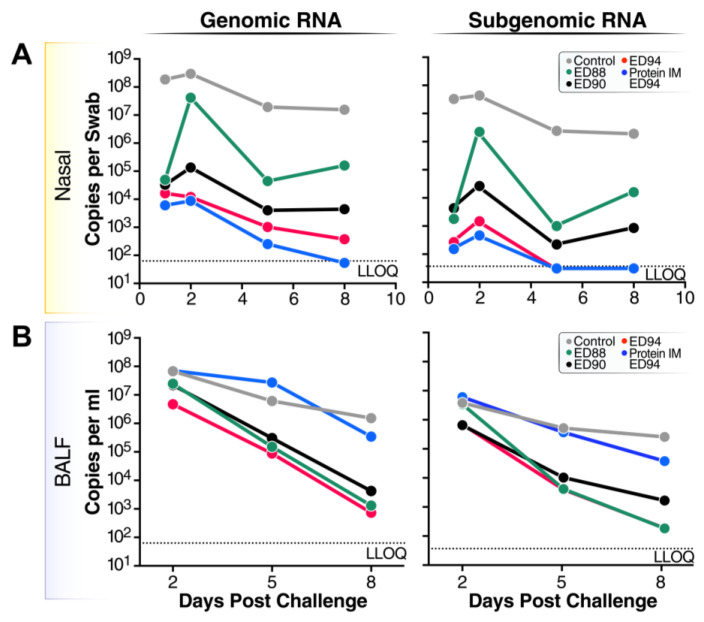
Viral loads are dramatically reduced in nasal mucosa and lower airways of vaccinated animals after challenge with SARS-CoV-2 Beta variant. Genomic and subgenomic mRNA was quantified by qPCR from (**A**) nasal swabs on days 1, 2, 5, and 8 post challenge or (**B**) BALF on days 2, 5, and 8 post challenge. Groups include vehicle control animals (grey circles, *n* = 5), animals vaccinated with ED88 (green circles, *n* = 6), ED90 (black circles, *n* = 6), intramuscular prime of spike protein followed with ED94 boost (blue circles, *n* = 4), and ED94 (red circles, *n* = 6). Data expressed as mean for each group; copies per swab for nasal mucosa, and copies per mL for BALF. Abbreviations: BALF, bronchoalveolar lavage fluid; LLOQ, lower limit of quantitation; qPCR, quantitative polymerase chain reaction.

**Figure 6 vaccines-12-00132-f006:**
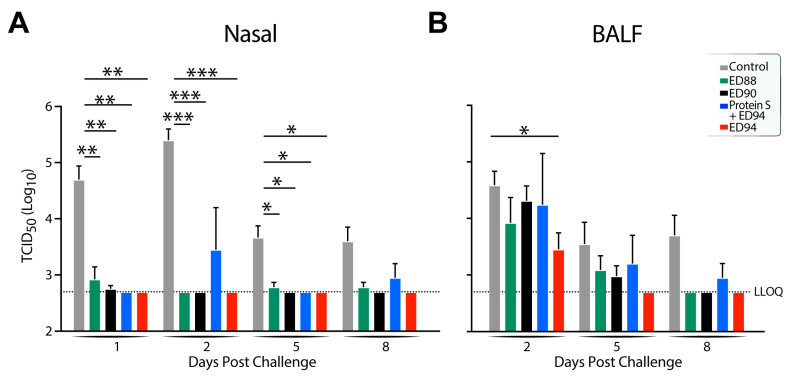
Shedding of SARS-CoV-2 in vaccinated animals is significantly reduced in upper and lower airways. Infectious SARS-CoV-2 Beta variant virus shedding was quantified by TCID_50_ using Vero cells and calculated using the Reed–Muench formula. (**A**) Nasal swabs collected on days 1, 2, 5, and 8 post viral challenge (**B**) BALF collected on days 2, 5 and 8 post viral challenge. Vehicle control animals (grey bars, *n* = 5), groups immunized with ED88 (green bars, *n* = 6), ED90 (black bars, *n* = 6), intramuscular prime of spike protein followed with ED94 boost (blue circles, *n* = 4), and ED94 (red bars, *n* = 6). LLOQ = 2.7. Data expressed as log10 TCID50/mL, mean ± SEM. Abbreviations: BALF, bronchoalveolar lavage fluid; LLOQ, lower limit of quantitation; TCID_50_, tissue culture infection dose 50.

**Figure 7 vaccines-12-00132-f007:**
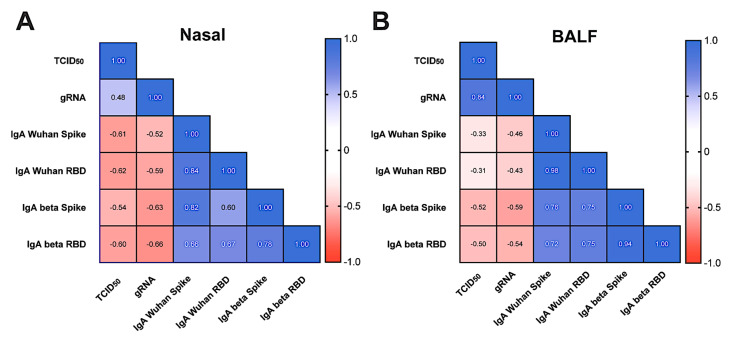
Vaccine-stimulated mucosal antibody production in the nasal passages and BALF negatively correlates with SARS-CoV-2 viral loads post challenge. Spearman’s r cross-correlation matrix heat maps comparing day 54 mucosal antibody responses to viral genomes and shedding 2 days post infection. (**A**) Nasal mucosal IgA against Wuhan or Beta full-length spike and RBD via surrogate virus neutralization tests compared to TCID50 and qPCR from nasal samples (**B**) BALF mucosal IgA against Wuhan or Beta full-length spike and RBD compared to TCID50 and qPCR from BALF samples. Two-tailed analysis; >0.5 or −0.5 indicates significant correlation; dark blue, 1 = strong positive correlation, dark red, −1 = strong negative correlation. Abbreviations: BALF, bronchoalveolar lavage fluid; gRNA, genomic RNA; IgA, immunoglobulin A; qPCR, quantitative polymerase chain reaction; TCID50, tissue culture infection dose 50; RBD, receptor binding domain.

## Data Availability

The data presented in this study are available on request from the corresponding author.

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
