# Peer review of "A Next-Generation Adenoviral Vaccine Elicits Mucosal and Systemic Immunogenicity and Reduces Viral Shedding after SARS-CoV-2 Challenge in Nonhuman Primates"

_vaccines, 2024, doi:10.3390/vaccines12020132_

Round 1

Reviewer 1 Report

Comments and Suggestions for Authors

The paper by Tedjakusuma et al conveys novel information on mucosal Ad5 based vaccines with SARS-COV2 spike.  The study is well designed and the immunological responses and viral load responses are robust.  There are some areas where the paper could be improved.

1.     Did the authors asses nasal CD8+ TRM generation? As the Kohlmeier lab has recently shown in Nature that these responses can limit viral load as well as transmission in a Sendai virus model.

2.     The authors need to discuss this in the context of the recent Barouch NHP paper in the same December issue of Nature.

3.     The authors need to discuss limitations of Ad5 in populations that have pre-existing immunity

Reviewer 2 Report

Comments and Suggestions for Authors

In this manuscript titled “An adenoviral vaccine candidate elicits mucosal and systemic 2 immunogenicity and reduces viral shedding after SARS-CoV-2 challenge in nonhuman primates”, the authors developed three unique recombinant non-replicating adenovirus serotype 5 (rAd5) SARS-CoV-2 vaccines and tested their efficacy in non-human primates - Africa Green Monkey. Various assays were performed to interrogate their functionalities against viral challenge post administration of the vaccines. The results demonstrated that the rAd5 vaccines are able to generate strong immune reaction against both homologues or cross-relative viral strains and can significantly reduce the viral loading and shredding after infections, illustrating a potentially novel method to delivery vaccines in response to the future SARS-CoV-2 infections. The whole manuscript is well-written with good logic and results are delivered clearly. Below are my questions or concerns.

1.      The resolution of the figures in main manuscript is low. Can this be adjusted? The supplementary figures are good with resolution.

2.      The vaccine design in figure 1B looks clear. However, the rational for the design is not well illustrated in the main manuscript (line 200-204). The authors might have clear description in their previous publications. But it would be great to include some major design concepts here for the reader to understand. To this point, I personally would like to know what is the function of the dsRNA adjuvant (how this double-stranded RNA works)? I do not see many explanations in the manuscript. Why choose Beta Actin promoter to expression nucleocapsid instead of CMV promoter? Are there any other structures other than nucleocapsid could be part of the vaccine? I am wondering if the authors would come up with a design principle for building this type of vaccine which could directly be used by the other researchers.

3.      Following point #2, based on the results, can the authors make a conclusion that design ED88 is over-performance than design ED90? If not, in which perspectives the nucleocapsid would enhance the vaccine efficacy, or is it necessary to incorporate this part to the vaccine? (Additional part might increase cloning steps/time and vector load for the vaccine)

4.      The connected scatter plot in figure 1 and 2 is a little confusing to me. It makes me feel like the fold-change among each time point is linear which I don’t think it is the case. In reality. Scatter plot without the connected line or even the bar chart (similar to figure 3) could be better.

5.      The authors claimed that intramuscular injection did not product serum neutralizing antibody by day 28, neither for the ED94 booster which did not increase its functional activity further by SVNT assay (line 299 - 302). But the PRNT assay does show great antibody titers in this group, even more than group ED88 and ED90 (figure 3b, blue bar). Why is that?

6.      In line 310, the authors claimed that 60% of animals primed and boosted with ED88 or ED90 also had protective serum IC50 titers. How does the 60% being calculated?

7.      Can you describe the rationale of using ACE-2 for the neutralization test in the manuscript and add any citations on it?

8.      I am curious that does the intranasal immunization-based vaccine could cause innate immune response? If so, how much would this affect the result?

9.      mRNA vaccine is also a type of functional vaccines which shown great success against SARS-CoV-2 infections. The authors have a few discussions about mRNA vaccine in the discussion section (line 513 - line 516). I am wondering if the authors have compared their results with previously mRNA vaccine data against SARS-CoV-2 virus and its variants. A little bit more discussion would be favorite.

10.   The BALF collected date may be inconsistent between line 122-123 and 133. Maybe I missed but I don’t see post challenge day1 BLAF data. If so, please correct.
